# Performance of a Genetic Algorithm for Estimating DeGroot Opinion Diffusion Model Parameters for Health Behavior Interventions

**DOI:** 10.3390/ijerph182413394

**Published:** 2021-12-20

**Authors:** Kara Layne Johnson, Jennifer L. Walsh, Yuri A. Amirkhanian, Nicole Bohme Carnegie

**Affiliations:** 1Department of Mathematical Sciences, Montana State University, Bozeman, MT 59717, USA; nicole.carnegie@montana.edu; 2Medical College of Wisconsin Center for AIDS Intervention Research, Milwaukee, WI 53202, USA; jwalsh@mcw.edu (J.L.W.); yuri@mcw.edu (Y.A.A.)

**Keywords:** DeGroot model, opinion diffusion, social influence, genetic algorithm, parameter estimation, social network intervention, pre-exposure prophylaxis (PrEP)

## Abstract

Leveraging social influence is an increasingly common strategy to change population behavior or acceptance of public health policies and interventions; however, assessing the effectiveness of these social network interventions and projecting their performance at scale requires modeling of the opinion diffusion process. We previously developed a genetic algorithm to fit the DeGroot opinion diffusion model in settings with small social networks and limited follow-up of opinion change. Here, we present an assessment of the algorithm performance under the less-than-ideal conditions likely to arise in practical applications. We perform a simulation study to assess the performance of the algorithm in the presence of ordinal (rather than continuous) opinion measurements, network sampling, and model misspecification. We found that the method handles alternate models well, performance depends on the precision of the ordinal scale, and sampling the full network is not necessary to use this method. We also apply insights from the simulation study to investigate notable features of opinion diffusion models for a social network intervention to increase uptake of pre-exposure prophylaxis (PrEP) among Black men who have sex with men (BMSM).

## 1. Introduction

Leveraging social influence is an increasingly common strategy to change population behavior or acceptance of public health policies and interventions. For example, an ongoing study—with a completed pilot—seeks to assess the feasibility of increasing pre-exposure prophylaxis (PrEP) interest for Black men who have sex with men (BMSM) through the use of a social network intervention: engaging and training network leaders to communicate the benefits of PrEP within their social networks [1]. Since the intervention is inherently a social network intervention based on the premise that the network leaders will be more influential than other network members or *agents*, an assessment of the intervention should incorporate both network structure and varying influence.

In order to analyze data from this study, we developed a novel genetic algorithm to fit DeGroot opinion diffusion models, incorporating the network structure and allowing for varying influence between agents in the network [2]. Fitting this model makes possible both predicting future opinions and interpreting parameters that describe the opinion diffusion process, allowing us—for example—to identify particularly influential or stubborn agents. We previously demonstrated algorithm performance across a variety of network and dataset features; however, the performance was assessed under relatively ideal conditions, ignoring common issues present in this and other public health applications. Informed by known limitations of the dataset and those expected in similar research, we perform a simulation study to assess the performance of the algorithm in the presence of ordinal (rather than continuous) opinion measurements, network sampling (not observing the full network), and model misspecification, providing researchers necessary information about the performance of this method under the assumption violations expected for health behavior interventions.

Given the ubiquity of Likert or other ordinal scales in social and behavioral science research, we expect most studies using opinion data will use an ordinal scale as is done in the PrEP study [3]. Since our selected model assumes opinions are continuous, we make the common assumption that ordinal data posses interval properties and convert them to a continuous [0,1] scale. This induces measurement error since the latent (continuous) opinions can only be measured as pre-defined values on the interval, determined by the number of items in the ordinal scale. Since both the objective function we optimize to fit a model and assessments of model fit incorporate the difference between modeled and observed opinions on the ordinal scale, the precision of the ordinal scale affects not only the quality of data provided to the algorithm but also the informativeness of each of these measures, making the use of ordinal data a concern for both model fitting and assessments of the usefulness of the model, in terms of parameter recovery, modeling observed opinions, and predicting future opinions.

The DeGroot model also assumes that the full network is sampled and the presence or absence of each possible link between agents is known. In practice, it is impractical to obtain the full social network due to agents declining to participate or failing to meet eligibility criteria and limitations of network sampling methods, such as the *snowball sampling* method used in the PrEP study where agents recruit other agents. While it may initially seem reasonable to determine the presence or absence of each possible link between sampled agents—particularly for the small networks for which this method was developed—by simply asking each agent if they know the others, our focus on public health applications introduces ethical concerns. When these networks are comprised of agents sharing stigmatized characteristics relating to sexuality, high-risk behaviors, or health status, identifying someone as a member of one of these networks is tantamount to revealing sexuality, high-risk behaviors, or health status. For these reasons, we expect missing agents and unknown links to be concerns for most applications of this method.

Finally, we consider the possibility that the opinion diffusion process does not follow the DeGroot model but *bounded confidence* and *decay* extensions to this model. Bounded confidence models are based on the premise that agents with substantially different opinions on a topic will either not discuss the topic or will be unable to influence each other if they do. This is of particular concern for the PrEP study since the network leader intervention is used to overcome disinformation and negative PrEP stereotypes within the social networks. Decay models allow for agents to be initially open to influence but become progressively more confident in their own opinions and less susceptible to influence over time. In the context of the PrEP study, this would allow for the intervention to initially be more effective, with the network leaders presenting new information using the techniques learned in training, but become less effective in the absence of any additional information after agents develop opinions based on the initial new information.

Since ordinal data and network sampling are concerns for the PrEP study and are expected for any other studies involving opinion diffusion on social networks, it is necessary to understand how the algorithm performs under these conditions. We also consider the possibility that the opinion diffusion process follows extensions of the simple DeGroot model for which the algorithm was developed: bounded confidence and decay. We detail these assumption violations along with the simulation study, the methodological development, and relevant details of the PrEP study in Section 2. In Section 3, we assess the ability of the algorithm to estimate parameters, model observed opinions, and predict future opinions. To provide concrete examples of the phenomena demonstrated in the simulation study, we revisit selected results from the PrEP study in Section 4. Finally, to facilitate researchers using this method, we include recommendations for using model fit to assess performance and dealing with with ordinal data, network sampling, and alternate models in Section 5.

## 2. Materials and Methods

We outline our methodological development, focusing on the aspects most relevant to the simulation study and its use on datasets known to violate model assumptions. To place the simulation study and discussion in context, we describe the PrEP pilot study, including how the networks were recruited, the intervention, data collection, and scales selected for analysis. Finally, we detail the assumption violations considered in the simulation study, the procedure, and performance metrics.

### 2.1. Modeling Opinion Diffusion

In this section, we first highlight features of the DeGroot model relevant to the simulation study. In particular, we focus on the necessary modifications to use the model with ordinal data. We also provide a brief overview of the development of the genetic algorithm and its application to opinion diffusion.

#### 2.1.1. DeGroot Model

The DeGroot model is the foundational opinion diffusion model and most influential non-Bayesian model [4,5,6,7]. Under this model, agents update their opinions as a weighted average of their current opinions and the opinions of their network contacts. This process is described on a network of *N* agents by
(1)X(t+1)=WX(t),
where X(t) is a vector of length *N* with $x_{i}\left(t\right)\in \left[0,1\right]$ representing the opinion of agent *i* at time *t* and *W* is an N×N matrix with $w_{ij}$ representing the weight that agent *i* places on the opinion of agent *j*. The elements in the weight matrix *W* are subject to the constraints $0\leq w_{ij}\leq 1$ and $\sum_{j=1}^{N}w_{ij}=1$, allowing $w_{ij}$ to be interpreted as the proportion of the total influence on agent *i* exerted by agent *j*. The weight matrix is further restricted based on the adjacency matrix *A*: an N×N matrix where $a_{ij}=a_{ji}=1$ if agents *i* and *j* have the potential to directly influence each other and $a_{ij}=a_{ji}=0$ otherwise. Though atypical in network analysis, we include self-links in the adjacency matrix ($a_{ii}=1$) so that agents are influenced by their current opinion during the update process. We subject the weight matrix to the constraint $w_{ij}\leq a_{ij}$ so that the weight matrix contains structural zeros where direct influence is not possible.

#### 2.1.2. Transformations

While the DeGroot model uses continuous opinions on the interval [0,1], in practical applications, opinions are typically measured using a Likert or similar ordinal scale and potentially combined into a composite scale. While interval properties are not inherent to ordinal data, it is a common assumption that allows for the application of mathematical operations and is implicit in the use of a composite scale. In order to use ordinal data with this model, we treat them as discrete, assuming they possess interval properties. Based on this assumption, we transform ordinal data to the continuous scale and back-transform the continuous opinions to the ordinal scale using the following process:

##### Forward Transformation:

Begin with data on an *n*-point ordinal scale, converting to a 1 to *n* scale if necessary.Divide the interval [0,1] into *n* sub-intervals of equal width.An opinion of *x* on the ordinal scale takes on the middle value, *y*, in the *x*th sub-interval on the continuous scale.

##### Back Transformation:

Begin with data on a continuous [0,1] interval to be converted to an *n*-point ordinal scale.Multiply the continuous opinion *y* by *n*.Round the multiplied continuous opinion up to an integer (ceiling function) to produce an opinion on the ordinal scale. (This final step does not work for the edge case where y=0, so any such values are automatically converted to an ordinal value of 1.)

#### 2.1.3. Objective Function

When using ordinal data, our goal is to find the parameters that best model the observed opinions on the ordinal scale. To this end, we minimize an objective function that penalizes deviation between observed and modeled opinions on the continuous scale only if they also differ on the ordinal scale according to
(2a)$$f\left(\hat{X},X\right)=\sum_{i=1}^{M}\sum_{t=0}^{T-1}B\left({\hat{x}}_{i}\left(t\right),x_{i}\left(t\right)\right)|{\hat{x}}_{i}\left(t\right)-x_{i}\left(t\right)|,$$
where *M* is the number of sampled or recruited agents and $B({\hat{x}}_{i}\left(t\right),x_{i}\left(t\right))$ measures the absolute deviation between the observed and modeled opinions on the ordinal scale. We refer to modeled opinions where $B({\hat{x}}_{i}\left(t\right),x_{i}\left(t\right))=0$ as being in the correct *bin* or as a correctly modeled opinion. This objective function can also be assessed on a row or agent level by excluding the sum across agents:
(2b)$$f\left({\hat{x}}_{i},x_{i}\right)=\sum_{t=0}^{T-1}B\left({\hat{x}}_{i}\left(t\right),x_{i}\left(t\right)\right)|{\hat{x}}_{i}\left(t\right)-x_{i}\left(t\right)|.$$

#### 2.1.4. Genetic Algorithm

We developed a genetic algorithm using selection, blending, crossover, mutation, and survival operators to fit the DeGroot model on opinion data. We define a chromosome as the weight matrix *W* and a gene as a single row of the weight matrix ($W_{i}$), representing the influence on agent *i*. Further details on the algorithm, implemented in Julia, and a preliminary performance simulation study are available in Johnson et al. [2,8].

### 2.2. PrEP Pilot Study

Since the network sampling method and scales used inform the simulation study, we provide an overview of the motivation and detail the network recruitment process and the measures selected for the analysis: self-efficacy and willingness. We also explain how the intervention was implemented and the relationship with the data collection process to provide the context necessary to understand our discussion of the models fit using these data. Lastly, we highlight the limitations of the social network information that can reasonably be collected to inform our decisions for the simulation study.

#### 2.2.1. Motivation

Pre-exposure prophylaxis (PrEP) greatly reduces risk for HIV acquisition and is widely recommended to be used by persons who engage in high-risk behaviors. HIV prevalence and incidence rates among Black men who have sex with men (BMSM) in the U.S. remain the highest when compared to any other group, making BMSM a key priority group for using PrEP; however, PrEP uptake by BMSM remains challenging. Negative PrEP-related stereotypes are common, while PrEP awareness is low among BMSM—particularly those who do not identify themselves as gay—and messages directed to gay community members may not reach them or be seen as personally relevant [9,10,11].

Social networks—structural elements of a community—carry an important functional utility for their members. They may provide an environment for mutual support and the exchange of trusted information, among other functions. Past research has primarily studied BMSM networks as mechanisms of HIV transmission; nonetheless, networks can be harnessed for interventions to achieve health-related goals, including HIV prevention [12]. Networks have been studied in the context of interventions among persons who inject drugs and to promote condom use in a variety of community populations, including men who have sex with men (MSM) [13,14,15].

Recent research is beginning to utilize networks to promote PrEP through increasing PrEP awareness, correcting PrEP misconceptions, and strengthening norms, attitudes, benefit perceptions, and skills for PrEP use. Recommendations from network leaders who are personally known and trusted are likely to be personally salient and may have greater impact than generic, impersonal messages. Peers can potentially be trained to endorse PrEP acceptance and counsel their network members in its benefits [16].

The intervention model is also grounded in principles of innovation diffusion theory [17]. After recruiting networks of BMSM in the community, this study selected a cadre of members within each network who were most socially interconnected with others, most trusted for advice, and most open to PrEP. These network leaders attended sessions where they learned about PrEP and its benefits, and were systematically engaged to talk with friends about these topics, correct misconceptions and counter negative stereotypes about PrEP, instill interest in PrEP, and guide interested friends in accessing PrEP providers. Thus, the intervention engaged trusted and socially-interconnected network leaders to function as agents to diffuse messages to others.

#### 2.2.2. Network Recruitment

This pilot intervention study was conducted in the period 2016–2017 in Milwaukee, WI with five distinct social networks of BMSM enrolled. In order to sample networks of BMSM, we employed a network enrollment method known as *snowball sampling*. Each social network was recruited by first identifying and enrolling *seeds*: members of the BMSM community who were located in venues such as clubs, hangout places, and drop-in centers for racial minority LGBT youth. Staff approached and invited the seed by introducing this study and screening to ensure that the seed met eligibility criteria: being assigned as male at birth; identifying as African American, Black, or multi-racial; being age 18 or older; reporting sex with males in the past year, and not having knowledge of being HIV-positive.

Upon enrollment, seeds identified their BMSM friends by first name or initials and were asked to give each friend a study invitation packet. Those persons who responded to the seed’s invitation were also screened for eligibility and enrolled, concluding the first *wave* of recruitment and establishing the first *ring* of network members surrounding the seed. To recruit the second ring of network members, first ring network members identified their own BMSM friends and were asked to share invitation packets with them. The interested second-wave network members were also screened for eligibility and enrolled. The five recruited networks of the seeds—each recruited using two waves—had a total of 40 members, with networks composed of between four and twelve participating members. Entry criteria for network members were the same as criteria for seeds except we did not restrict study eligibility based on network member serostatus.

#### 2.2.3. Intervention and Data Collection

Network leaders were selected to attend a group intervention providing PrEP education and skills training in how to endorse PrEP to friends. This intervention met for two hours per session each week for five weeks. All participants in this study (network leaders and other network members) completed assessments at enrollment and 3 months later, following the group intervention with network leaders. Assessment measures were completed by computer using self-administered questionnaires during individual sessions at the time of the baseline and follow-up visits. Further information on procedures is available in Kelly et al. [1].

#### 2.2.4. Measures

Key measures for this analysis were PrEP self-efficacy and PrEP willingness. Self-efficacy was selected as an outcome due to its important role in health behavior theories such as the theory of planned behavior and the information-motivation-behavioral skills model [18,19]. Self-efficacy predicts health outcomes, and meta-analysis has shown that experimentally-induced changes in self-efficacy predict future behavior [20,21]. PrEP self-efficacy specifically has been shown to correlate with intentions to use PrEP and PrEP use among MSM and with willingness to use PrEP among people who use drugs [22,23,24,25]. Similarly, in the pilot context, willingness was selected as an outcome expected to precede later behavior (PrEP uptake). Willingness to use PrEP has been a key focus of the literature as PrEP has emerged as a new prevention tool and was a target of the social network intervention [1,26,27,28,29]. Willingness has been shown to correlate with health behavior [30,31].

Scales assessing PrEP self-efficacy and willingness were drawn from the literature [22]. PrEP self-efficacy was assessed with eight items (Cronbach’s α=0.70). Each item asked participants to use a 4-point scale to indicate how difficult, from very hard to very easy, it would be to engage in an action (sample item: “How difficult or easy would it be for you to visit a doctor who can provide PrEP?”). PrEP willingness was assessed with three items (α=0.81). Each item asked participants to indicate their strength of agreement using a 5-point Likert scale, from “strongly disagree” to “strongly agree” (sample item: “I would be willing to go on PrEP if I had a casual sex partner who was HIV-positive.”). Scale scores for both constructs were created by summing items, resulting in a 25-point scale for self-efficacy (8–32) and a 13-point scale for willingness (3–15).

#### 2.2.5. Limitations

Despite its numerous advantages, network research presents certain challenges and complexities. In particular, research on sensitive topics—or when research is being undertaken among vulnerable population members—requires caution over the information being collected, and ethical considerations sometimes prevent collection of certain types of data. In HIV prevention research, network data collection is often limited to prevent unintentionally revealing the HIV status or sexual orientation of members within the same network to one another. For example, in MSM networks or in networks of people living with HIV infection, assessment of ties between two network members could reveal the HIV status or sexual identity of one network member to another. This can be ethically unsound because members of an *ego* or seed’s network may be unaware of one another’s MSM or HIV-positive status prior to this assessment. Thus, simply revealing someone as a member of the ego’s MSM network may lead to disclosing that person’s sexual orientation. These ethical considerations result in sampled networks with unknown links between agents for studies on health interventions or other sensitive topics.

Another challenge is associated with completeness of the network data that can feasibly be collected. While complete network data requires the enrollment of all members from a given network, it can rarely be achieved for numerous reasons: a seed’s inability to pass an invitation packet or encourage the friend’s enrollment; a friend’s lack of interest, time available to complete study procedures, or willingness to participate; a network member not meeting eligibility criteria; dropping out from a network during this study; and staff inability to contact a person using the available information and methods. This results in not only the individual who could not be enrolled being missing from the sampled network but also a break in the recruitment chain that would have been produced through that person. Additionally, the sampling method automatically results in agents who are part of the third ring or beyond being missing from the sampled network.

### 2.3. Simulation Study

The previous simulation study demonstrated algorithm performance across a variety of network and dataset conditions but did not address expected problems with real-world datasets, so we conduct a follow-up simulation study under the less-than-ideal conditions likely to arise in practical applications. We focus in particular on the PrEP study to provide context for the models fit using these data, addressing ordinal data, network sampling, and model misspecification. We also present the simulation study inputs relevant to these assumption violations: scale, recruitment probability, adjacency matrix variety, and bounded confidence and decay parameters. Finally, we detail the procedure for conducting the simulation study and the performance metrics.

#### 2.3.1. Ordinal Data

Under the assumption that latent opinions are on a continuous [0,1] scale, measuring opinions on an ordinal scale induces measurement error. Since we convert ordinal data to the continuous scale according to Section 2.1.2, an *n*-point ordinal scale results in more precise opinions on the continuous scale for larger values of *n*. We consider ordinal scales with 5, 7, 10, 20, and 30 points. These are selected based on the scales used for the PrEP study and typical ordinal or Likert scales, with the more precise scales intended to represent composite scales. To be consistent with the model, we assume latent opinions are continuous and that these continuous opinions are shared with network contacts without error.

#### 2.3.2. Adjacency Matrix

Another assumption of the selected model is that all agents in the network are sampled and all links between these agents are known. Since the PrEP study uses *snowball sampling*—where agents recruit other agents—with two recruitment waves, we expect that agents are missing from the sampled network. While these missing agents do result in missing links to those agents, we are also interested in the effect of agents who are included in the sampled network but where the presence or absence of a link to another sampled agent is unknown.
**Missing Agents:** Since this study uses two waves, agents with a *geodesic distance* to the seed larger than two—those who are friends of friends of friends of the seed or further removed—are always excluded from the sample. Additionally, some nominated agents may decline to participate, with 53% of the individuals named during the recruitment process agreeing to participate in the PrEP pilot study. We consider both the possibility of guaranteed recruitment (p=1) and non-guaranteed recruitment (p=0.5), informed by the recruitment percentage in the pilot study.**Unknown Links:** While we have described situations where the presence or absence of a link in the network is unknown, the adjacency matrix does not have the flexibility to indicate an unknown link, requiring us to specify either the presence ($a_{ij}=a_{ji}=1$) or absence ($a_{ij}=a_{ji}=0$) of a link between agents *i* and *j*. When information about all nominations—including repeats—is available, we know all links between the sampled agents. We refer to the adjacency matrix where all links between sampled agents are known as the *correct* matrix. Note that “correct” refers only to the links between sampled agents and does not imply all agents in the true network are included in the sample. Unfortunately, it is usually impractical or unethical to obtain the information necessary for the *correct* matrix. We include it in the simulation study, not as a viable solution to the issue of missing links, but as a baseline for comparison for more practical solutions and to determine the consequences of failing to collect the information necessary for the *correct* matrix.

In cases where only the the nominations leading to initial recruitment are available, links between agents in the same wave or between the first and second wave are missing from the sampled network. We refer to the adjacency matrix where only recruitment links are recorded as the *build* matrix since we begin with a matrix of zeros and add only links known to exist. While the information for the *build* matrix can be easily and ethically obtained using any sampling method relying on nominations, it has the potential to negatively affect estimation. When the link between agents *i* and *j* is excluded from the sampled network, $w_{ij}$ and $w_{ji}$ are structurally zero, making it impossible for the algorithm to identify any influence between agents *i* and *j*. In contrast, if the link is included in the sampled network, the algorithm can identify a solution where ${\hat{w}}_{ij}={\hat{w}}_{ji}=0$, meaning it is possible for the algorithm to identify the absence of influence between agents *i* and *j*.

To avoid the estimation problems inherent in the *build* matrix, we propose the *remove* matrix, beginning with a matrix of ones and remove any links known to be absent. In the context of snowball sampling, we assume the seed is not linked to any agents beyond those he names, but any links for agents between or within waves could potentially exist. Given the promising results using the *remove* matrix for addressing both unknown links and missing agents, we also consider the *complete* matrix—where all sampled agents are assumed to be linked—resulting in a matrix of ones. See the discussion relating to network 5 in Section 4 for information on why including a link known not to exist, as is done with the *complete* matrix, has the potential to improve performance.

#### 2.3.3. Model Misspecification

While we have only implemented the algorithm for the DeGroot model, we consider the possibility that the opinion diffusion process instead follows *bounded confidence* and *decay* extensions to this model. This allows us to assess whether the current version of the algorithm is useful when bounded confidence and decay are expected. Since the presence of bounded confidence and decay are not mutually exclusive, we include cases where both are present.
**Bounded Confidence:** For bounded confidence, we assume agents with sufficiently differing opinions will either not discuss the topic or will be otherwise unable to influence each other. This is accomplished through the addition of the restriction on the weight matrix *W*
(3a)$$w_{ij}=w_{ji}=0\;\mathrm{if}\;\left|x_{i}\left(t\right)-x_{j}\left(t\right)\right|>\Delta$$
where Δ∈(0,1] represents the maximum difference between opinions after which agents are unable to influence each other [32]. This is equivalent to the DeGroot model when Δ=1. Since changing opinions allow for agents falling within the threshold for bounded confidence at some time steps and not at others, the application of bounded confidence necessitates a notation adjustment. We define *W* as the weight matrix in the absence of bounded confidence restrictions and W(t) as the weight matrix after applying the appropriate bounded confidence adjustments at time *t*. Based on this new notation, we update to
(3b)$$w_{ij}\left(t\right)=w_{ji}\left(t\right)=0\;\mathrm{if}\;\left|x_{i}\left(t\right)-x_{j}\left(t\right)\right|>\Delta .$$

The changing weight matrix also means the weight matrix W(t) may not meet the sum-to-one constraint after the application of Equation (Equation 5). To correct this, any non-zero weights $w_{ij}$ and $w_{ji}$ must be redistributed within the row when $|x_{i}\left(t\right)-x_{j}\left(t\right)|>\Delta$. We redistribute the weight proportionally using
(4)$$W_{i}\left(t\right)=\frac{W_{i}\left(t\right)}{1-\sum_{j∋|x_{i}\left(t\right)-x_{j}\left(t\right)|>\Delta }w_{ij}}.$$As these models restrict influence to those with similar opinions, bounded confidence reduces the potential for agents to change their opinions. Since this reduction in potential opinion change is most severe when Δ is further from one, we consider data generated under bounded confidence models with bounded confidence parameter Δ of 0.1, 0.5, and 0.9.
**Decay:** The second extension of the model allows for agents to place changing weight on their own current opinion according to
(5a)$$X\left(t+1\right)=\left(\left(1-\lambda_{t}\right)I+\lambda_{t}W\right)X\left(t\right)$$
where *I* is an N×N identity matrix and $\lambda_{t}\in \left(0,1\right]$ is a scalar adjustment factor allowed to vary with time [32]. The effect of the adjustment is to shift the weight for each agent to (or from, depending on the values of $\lambda_{t}$) their self-weight. In order for such a model to be useful, we impose a structure: setting $\lambda_{t}$ equal to λ to the power of *t*, so that agents place decaying weight on the opinions of others. We modify the previous equation to
(5b)$$X\left(t+1\right)=\left(\left(1-\lambda^{t}\right)I+\lambda^{t}W\right)X\left(t\right)$$
with λ∈(0,1]. Equation (Equation 8) is equivalent to the DeGroot model if λ=1. Given the structure imposed on $\lambda_{t}$, these models result in an opinion diffusion process where agents place progressively more weight on their own opinions over time, resulting in more confident or stubborn agents whose opinions change less with each time step. This effect is most pronounced when λ is further from one, so we consider data generated under decay models with decay parameter λ of 0.1, 0.5, and 0.9.


#### 2.3.4. Procedure

Table 1 summarizes the inputs used in the simulation study which we implement in Julia [8]. The hyperparameters used are the same as in Johnson et al. [2]. We consider all possible combinations of the inputs in the table and replicate every combination ten times. We begin by generating an Erdős–Rényi network (while the lack of clustering in these networks is not reflective of the structure of larger networks of BMSM, the generated networks are intended to represent clusters from these large networks as the PrEP study targets these clusters; additionally, the structure of the network is incidental for agents with geodesic distances to the seed larger than three since these agents are unable to directly influence sampled agents.) of a specified size and target degree, rejecting any networks that are not connected (i.e., contain a path between any pair of nodes). We draw initial opinions (X(0)) from a Unif(0,1) distribution and randomly generate a weight matrix (*W*) with a target self-weight, using these to simulate opinions across an additional 20 time steps according to Equation (Equation 1). We then use the back-transformation process to convert all data to an *n*-point scale.

To simulate the snowball sampling process, we identify a seed using degree centrality. Each agent connected to the seed is then recruited with probability *p*. For each recruited agent, we repeat the process: recruiting each of their contacts with probability *p*. Since the probability of recruitment is the probability that an agent will agree to participate, this probability check is only applied once, regardless of the number of times each agent is nominated. Agents are excluded from the sampled network once they first decline to participate, and networks with less than four sampled agents are rejected.

We then create the appropriate adjacency matrix variety using the sampled agents. The generated adjacency matrix, reduced to include only the sampled agents, is the *correct* matrix, and the *complete* matrix is a square matrix of ones, with its size determined by the number of sampled agents. To create the *build* matrix, we remove all links between agents recruited in the same wave from the *correct* matrix and remove all but a randomly selected connection to a first-wave agent for each second-wave agent. Finally, we create the *remove* matrix by adding links between all agents in either the first or second wave to the *correct* matrix. We provide only the ordinal opinions for the sampled agents across the specified number of time steps and the appropriate adjacency matrix variety to the algorithm.

#### 2.3.5. Performance Metrics

When assessing algorithm performance, we are interested in its ability to do three things: recover the parameters used to generate the data, model the latent (continuous) opinions on the observed time steps (those provided to the algorithm), and predict future opinions (time steps past those provided to the algorithm). To assess parameter recovery, we use root-mean-square error (RMSE):(6)$$RMSE_{rec}=\sqrt{\frac{\sum_{i=1}^{M}\sum_{j=1}^{M}{\left(w_{ij}-{\hat{w}}_{ij}\right)}^{2}}{\sum_{i=1}^{M}\sum_{j=1}^{M}a_{ij}}}=\sqrt{\frac{\sum_{i=1}^{P}{\left(w_{p}-{\hat{w}}_{p}\right)}^{2}}{P}},$$
where *P* is the number of elements not fixed at zero in the weight matrix (the number of parameters to be estimated) and $w_{p}$ is the *p*th non-structurally-zero element, with $w_{p}$ and ${\hat{w}}_{p}$ representing the true and estimated weights, respectively. The adjacency matrix used to determine which values are structurally zero is the variety provided to the algorithm except for the *build* matrix where the *correct* matrix is used to penalize the incorrectly-identified structural zeros in the *build* matrix. Note that the weight matrix is also reduced to only sampled agents, meaning the rows may no longer sum to one due to weight placed on now-missing agents during the data generation process.

We also assess modeling opinions on observed time steps and predicting opinions past the observed time steps using RMSE. Recall that we generate 21 time steps, providing the first *T* time steps to the algorithm, and do not assess fit on initial opinions, as all opinions past initial are modeled based on the initial opinions. Modeling RMSE is assessed on the T−1 time steps past initial provided to the algorithm and prediction RMSE is assessed on the 21−T time steps past those provided to the algorithm according to
(7)$$RMSE_{mod}=\sqrt{\frac{\sum_{t=0}^{T-1}\sum_{i=1}^{M}{\left({\hat{x}}_{i}\left(t\right)-x_{i}\left(t\right)\right)}^{2}}{M(T-1)}}$$
and
(8)$$RMSE_{pred}=\sqrt{\frac{\sum_{t=T}^{20}\sum_{i=1}^{M}{\left({\hat{x}}_{i}\left(t\right)-x_{i}\left(t\right)\right)}^{2}}{M(21-T)}}.$$
Since none of these measures are available during practical applications of the algorithm, we also include the fit assessment that would be available to users: model fit using ordinal opinions or *ordinal fit*. This allows us to determine what ordinal fit on the observed time steps tells us about parameter recovery, modeling latent opinions, and predicting latent opinions. We again use RMSE:(9)$$RMSE_{fit}=\sqrt{\frac{\sum_{t=0}^{T-1}\sum_{i=1}^{M}B^{2}\left({\hat{x}}_{i}\left(t\right),x_{i}\left(t\right)\right)}{M\left(T-1\right)n^{2}}},$$
where *n* is the number of items in the ordinal scale.

## 3. Results

We assess the impact of network sampling, ordinal data, and model misspecification on performance in terms of parameter recovery, modeling latent (continuous) opinions, and latent opinion prediction. We also investigate the degree to which fit on observed time steps, the only performance measure available to the user, is indicative of our chosen performance metrics: recovery, modeling, and prediction. We relegate model misspecification to the subsection on alternate models and consider only data generated under the DeGroot model for the rest of the section since the alternate models are a potential extension—but not the primary purpose—of the algorithm and the results suggest they are of little concern. Note the differing y-axis scales within all plots.

### 3.1. Network Sampling

The snowball sampling approach with the potential for agents to decline to participate results in unknown links and missing agents. While the *build*, *remove*, and *complete* matrices focus specifically on handling these unknown links, we find that the *remove* matrix is useful for addressing both problems. Figure 1 assesses recovery, modeling, and prediction by adjacency matrix type and number of time steps. As expected based on its inflexibility, the *build* matrix consistently performs the worst across all measures. Since the *correct* matrix was included as a baseline for assessing more viable solutions to unknown links, its equivalent or slightly worse performance than the *remove* and *complete* matrices indicates not only that good solutions are available when there are unknown links in the dataset, but also that attempting to determine the status of these links would provide little to no benefit. The nearly identical performance of the *remove* and *complete* matrices indicates that, while including links in the adjacency matrix whose presence or absence in the true network is unknown is beneficial, there is no additional benefit to including links in the adjacency matrix known not to exist in the true network, especially since it complicates interpretation. For the above reasons, we use only the *remove* matrix for the remainder of the results.

### 3.2. Ordinal Data

The use of ordinal data—with an assumption that they possess interval properties—instead of the continuous opinions in the DeGroot model induces measurement error, with ordinal scales containing more points being more precise. Figure 2 assesses recovery, modeling, and prediction by number of items in the ordinal scale and number of time steps: the quality and quantity of information provided to the algorithm. For modeling and prediction, more precise ordinal scales improve performance, as does the use of more time steps. In particular, prediction benefits most strongly from additional time steps: switching from two to three time steps can result in more improvement than even a substantial increase in the precision of the scale. Recovery improves with more points in the scale for three or six time steps but worsens on only two time steps, suggesting possible overfitting.

### 3.3. Alternate Models

We determine whether this method—which assumes the opinion diffusion process follows the DeGroot model—could reasonably be used when the process actually follows bounded confidence and decay extensions to this model. Figure 3 assesses recovery, modeling, and prediction by decay parameter (λ) and bounded confidence parameter (Δ) with the “NA” level indicating the absence of either bounded confidence or decay. To provide context, extreme bounded confidence parameters (low values) result in little or no change in opinions and extreme decay parameters (low values) mean agents are initially open to influence but quickly become unwilling to change their opinions. Note that the the purple violin in the rightmost facet represents the DeGroot model. For recovery, there is very little concern for even extreme bounded confidence or decay parameters. There is a decrease in performance with lower decay parameters for both modeling and prediction, with much larger changes for prediction. In both cases, low values of the bounded confidence parameter moderate this effect since changing receptivity to influence is irrelevant in cases where extreme bounded confidence prevents influence from all but those with very similar opinions.

### 3.4. Performance Diagnostics

Since none of the performance metrics discussed above are available outside of the simulation study, we explore the extent to which the only measure available in practical applications—ordinal fit on observed time steps—is indicative of performance in terms of recovery, modeling, and prediction. Figure 4 shows the relationship between ordinal fit and performance in recovery, modeling, and prediction, accounting for precision of the ordinal scale and number of time steps. To better show small differences while accounting for zeros, we shift ordinal fit RMSE by 0.0001 and apply a log transformation. Note that (ordinal) model fit is a more informative measure for more precise scales since it is easiest to predict ordinal opinions when a single ordinal value covers a wider range of continuous opinions. Similarly, it is easier to identify a perfectly fitting model when there are fewer time steps on which opinions must be correctly modeled.

The vertical lines of points in the plot, which occur where the ordinal fit RMSE is zero, show runs where the model perfectly predicts ordinal opinions on the observed time steps. Based on these lines, it is clear there are local minima of the objective function that perfectly fit the data without recovering the parameters, particularly for less precise scales and fewer time steps. This means even perfect fit does not indicate good parameter recovery; however, poor fit does suggest poor parameter recovery. Unsurprisingly, ordinal fit best predicts modeling of latent opinions, particularly for more precise scales and more time steps. Perfect fit remains a poor indicator with less precise scales and fewer time steps but is meaningful with a precise scale, regardless of the number of time steps. Prediction is roughly the same as modeling except that perfect ordinal fit remains uninformative for fewer time steps even with precise scales.

## 4. Discussion

We revisit the results of the pilot PrEP study, placing them in the context of the simulation study. Only selected results are presented here, with full results available in Johnson et al. [2]. Since these models are fit using only two time steps and use a fairly precise scale, the estimates presented here are less accurate and conclusions relating to the PrEP study should instead be drawn based on an analysis of the data from the full study, once available. We reassess these results to provide a concrete example of how this method can be used and demonstrate some of the phenomena seen in the simulation study. This section is intended to serve only as an example and not as a comprehensive analysis of the PrEP data

All results presented are from ten separate runs of the algorithm for each adjacency matrix, network, and measure combination. Though we strongly discourage the use of the *build* matrix based on the results of the simulation study, both the *build* and *remove* matrices are used here for comparison. We also note the two observations per agent, taken three months apart, are treated as time steps t=0 and t=3, with time steps t=1 and t=2 missing, to allow for indirect influence, and that missing values on the observed time steps were imputed.

Figure 5 shows the deviation between observed and modeled opinions at follow-up across ten runs of the algorithm for willingness and self-efficacy using the *build* and *remove* matrices. While these deviations are measured in bins, note that self-efficacy uses a more precise 25-point composite scale while willingness uses a 13-point composite scale, so a bin covers a wider range of continuous opinions for willingness than for self-efficacy. Though we did find differences in performance across scales in the simulation study, these two measures differ in more than just the scales used, so we mention this only to highlight that a bin is not comparable between the two measures.

Figure 5 demonstrates the overall improvement in model fit for the *remove* matrix compared to the *build* matrix, but it also provides a specific example of why the *remove* matrix results in better model fit. The ten modeled opinions for the self-efficacy *build* matrix combination that are seven bins away from the observed opinions represent a single agent across the ten runs. We will refer to this agent as Ali and the other two agents involved in this example as Tom and Moe, with these names used for illustrative purposes only. Over the course of this study, Ali’s self-efficacy score increases, but his only connection in the *build* matrix is to Tom, whose initial self-efficacy score is lower than Ali’s. Consequently, the *build* matrix does not contain any connections that can explain the increase in Ali’s score, resulting in consistently poor estimates of Ali’s opinion at follow-up. When we use the *remove* matrix, Ali has potential connection to a variety of other agents including Moe, whose initial self-efficacy score is higher than Ali’s. Since the connection to Moe, and potentially to other agents, can now be used to explain the change in Ali’s score, we are better able to model Ali’s change in score. This is not to say that Ali is necessarily connected to Moe in the true network as we discuss in the following example.

Table 2 and Table 3 show the estimated weight matrices for network 5 with means across the ten runs using both the *build* and *remove* matrices for willingness and self-efficacy, respectively. We select this small network and exclude variability estimates for readability. Bold values in the *remove* matrix are structural zeros in the *build* matrix. Again, names are included purely for illustrative purposes. For these estimated weight matrices, we assess the relationships between Jay and Uba and Uba and Max, beginning with the estimated weight matrices for willingness. In the *build* matrix, we assume Jay and Uba are not connected but allow for the possibility of a link in the *remove* matrix. The average estimates of 0.00 in both directions of influence for the *remove* matrix suggest the absence of a link between Jay and Uba or at least the lack of influence. This also explains the minimal change in estimates for Jay between the two matrices.

There are, however, substantial changes in the estimates for Uba between the two matrices. When not fixed at zero, Uba’s influence on Max is still estimated to be 0.00, but Max’s influence on Uba is estimated as 0.32. There are two potential explanations for these seemingly contradictory estimates. It is possible that, though Uba and Max are connected in the true network, the nature of their relationship or beliefs about PrEP means Uba is influenced by Max, while Max does not value Uba’s opinion. Another reasonable explanation is that Uba and Max are not connected, but Uba is, instead, influenced by an agent missing from the sampled network whose willingness score is similar to Max’s. Table 3 suggests the latter explanation since neither Uba nor Max are influenced by the other for self-efficacy. This explanation shows how the *remove* matrix can improve modeling and prediction by allowing agents to place weight on, if not the correct agent, an agent with roughly the correct score.

It also demonstrates the potential for the less intuitive effect of improving overall recovery when agents are missing from the sampled network. Assuming the true weight Uba places on Max is a structural zero, an estimated value of 0.32 clearly contributes to incorrect recovery, but it also changes the other estimates in the row, hopefully bringing them closer to the true weight. If we assume Uba’s true self-weight is 0.68 and that Uba is uninfluenced by Eli, as estimated in the *remove* matrix, recovery RMSE for Uba goes from $\sqrt{\frac{{\left(0.00-0.04\right)}^{2}+{\left(0.68-0.96\right)}^{2}}{2}}=0.20$ for the *build* matrix to $\sqrt{\frac{{\left(0.00-0.00\right)}^{2}+{\left(0.68-0.68\right)}^{2}+{\left(0.00-0.32\right)}^{2}}{3}}=0.18$ for the *remove* matrix, ignoring that the weight placed on Jay was not structurally zero. (The weights for Uba that we are presenting as the ground truth for this example (0.00, 0.00, 0.68, and 0.00) do not sum to one without the weight of 0.32 placed on a missing agent. This is consistent with how we calculate recovery RMSE with missing agents in the simulation study. If we instead acknowledge that the true weight placed on Jay is a structural zero that is correctly estimated, the calculation is $\sqrt{\frac{{\left(0.00-0.00\right)}^{2}+{\left(0.00-0.00\right)}^{2}+{\left(0.68-0.68\right)}^{2}+{\left(0.00-0.32\right)}^{2}}{4}}=0.16$. Since the purpose of the example is to show that placing non-zero weight on a link not present in the true network can improve overall recovery, the inclusion of a correctly estimated structural zero obscures this point.) While the above example relies on the unreasonable assumption that the estimated weights perfectly match the true weights other than weight placed on Max, the direct impact on recovery within Uba’s row is not the only way the estimate of 0.32 improves recovery. The incorrect weight of 0.32 also produces more accurate modeled opinions for Uba, potentially improving weight recovery for any agents influenced by Uba. This process can also continue: improving the modeled scores of agents influenced by Uba which, in turn, improve recovery for their contacts.

Finally, we present the results comparing leaders to non-leaders that are most appropriate given the recommendation to use the *remove* matrix for parameter recovery. Table 4 shows the average weight placed on leaders and non-leaders and the difference between the two (leader−non-leader) across networks for willingness and self-efficacy, excluding self-weights and agents not connected to a leader. The higher mean weight for each leader and non-leader comparison is noted in bold. For willingness, leaders are consistently more influential, with the exception of network 4. The trend for self-efficacy is the opposite: non-leaders being more influential than leaders, with network 4 again being an exception. It is worth noting that both the mean weights and differences are typically lower in absolute value for self-efficacy than for willingness.

Since the behavior of network 4 is inconsistent with other networks in terms of the effect of the leader training intervention for both willingness and self-efficacy, this network merits additional assessment. Figure 6 shows the network representations of the *build* and *remove* adjacency matrices for network 4. Note that the *build* matrix represents the network as sampled—the links known to exist based on the recruitment chain—and the *remove* matrix represents the network provided to the algorithm. Again, names are only for narrative purposes. The agent in yellow is the seed, and the agent in green is the only agent in the network who attended leadership training. While other networks had agents other than the seed who attended training, network 4 is unique in having a seed who did not attend training. We include Table 5 with the estimated weight matrices for willingness and self-efficacy using the *remove* matrix for network 4 to support this example. Variability estimates are again excluded for readability.

The different behavior of network 4 suggests that a leader’s position in the network is important for a successful intervention; however, we must also consider the consequences for estimation of the only leader being recruited in the first wave of recruitment instead of as the seed. Specifically, the only leader in the network has unknown links and is directly linked to *peripheral agents*—those recruited in the second wave. These peripheral agents are expected to have links to the most missing agents: those who would theoretically have been recruited in a third wave. Since any weight placed on missing agents must be redistributed within the estimated weight matrix, an agent with links to more missing agents will have less accurate estimates for their row of the weight matrix. In this specific case, the only agent recruited in the second wave (Ray), places drastically differing estimated weights on the leader: 0.03 and 0.91 for willingness and self-efficacy, respectively. Given these extreme estimates, especially when the weight placed on others is typically lower for self-efficacy than for willingness, these estimates are likely being influenced by agents missing from the sampled network.

Regarding unknown links, the leader has potential links to both Cam and Obe, with the leader’s influence on Obe estimated to be 0.00 for both willingness and self-efficacy. Assuming these zeros indicate the absence of a link, these estimates, while correct, artificially decrease the average weight placed on the leader. Excluding zero or nearly zero estimated weights from the calculations in Table 4 is a potential solution, but this artificially inflates influence in cases where zero or nearly zero estimates are indicative of a failure to influence instead of the absence of a link. If this approach is used, recall that both $w_{ij}$ and $w_{ji}$ will be zero in the absence of a link between agents *i* and *j*. Using the median instead of the mean for summary statistics on estimated weights also has the potential to minimize the effect of correctly estimated structural zeros without requiring a decision on whether estimates indicate the absence of a link or failure to influence. While we have shown the *remove* matrix is the best solution to unknown links, this example highlights limitations when condensing estimates into summary statistics. We do not provide a specific recommendation but, instead, suggest assessing the options presented here with an awareness of the limitations and assumptions inherent in the approach selected.

## 5. Conclusions

We assessed the performance of the genetic algorithm for fitting DeGroot opinion diffusion models in terms of parameter recovery, modeling latent opinions, and predicting future opinions, considering known or expected problems of real-world datasets: ordinal data, network sampling, and alternate models. We also investigated whether the only performance metric available to the user, how well the model fits the data, is informative in terms of recovery, modeling and prediction. We highlight the results most relevant to researchers using this method when these assumption violations are known or expected.

Since even perfect fit is a poor indicator of parameter recovery, we recommend running the algorithm multiple times to identify a variety of solutions that produce perfect or very good fit. Note that averaging estimated weights across multiple runs will preserve the sum-to-one constraint. If multiple runs of the algorithm result in models with poor fit, the assumption violations are likely too extensive for use of this method. Good ordinal fit does suggest better modeling and prediction of latent opinions, particularly for more precise scales with more time steps. As with recovery, perfect fit should be viewed skeptically, especially with less precise scales and fewer time steps. This is especially true for prediction with only two time steps, regardless of the scale.

For alternate models, we considered *bounded confidence* models, where agents are not influenced by those with sufficiently differing opinions, and *decay* models, where agents become less open to the opinions of others over time. We found that even extreme values of the bounded confidence parameter are not concerning for recovery, modeling, or prediction. Decay models are also of little concern, especially when moderated by extreme bounded confidence parameters. We caution against the use of the model for opinion prediction when moderate to extreme decay is expected. Note that the potential for both bounded confidence or decay can be identified by looking at opinion data. Opinions that initially change quickly and become progressively more consistent suggest decay, and opinions that change minimally or not at all suggest bounded confidence. While little to no change could also be the result of a network comprised of very stubborn agents, this is not a particularly meaningful distinction as the algorithm handles data involving bounded confidence very well and the result is the same either way: agents place weight on only themselves or those with similar opinions.

A more precise ordinal scale—one with more points—generally improves recovery, modeling, and prediction with the exception of parameter recovery with only two time steps, where more precise scales result in slightly worse recovery. Consequently, we suggest prioritizing at least three observations per agent over a more precise scale when estimated parameters are of primary interest. When prediction is the primary goal, we recommend revisiting Figure 1 and the related discussion since more time steps can improve prediction more than a precise scale. In all other cases, using more precise scales should be considered as an alternative to collecting more observations per agent to improve overall performance with minimal impact to cost or participation.

While network sampling results in both agents missing from the sampled network and unknown links between sampled agents, the *remove* matrix—where links are included unless known not to exist—is a solution to both problems. We suggest using this matrix in most situations. It outperformed the *correct* matrix—containing correct information about all links between sampled agents—not just for modeling and prediction, but also for parameter recovery. When producing summary measures for parameter estimates using the *remove* matrix, we suggest revisiting the discussion relating to Table 5. Given the lack of benefit and impracticality, we do not recommend attempting to determine all links between sampled agents. Since the *complete* matrix—with links between all agents—had roughly equivalent performance to the *remove* matrix in terms of recovery, modeling, and prediction, we advise its use only when missing agents are expected and all links between sampled agents are known, making the *correct* matrix the only alternative. Finally, we strongly discourage the use of the *build* matrix—including only links known to exist—in all cases as it has consistently poor performance.

Overall, this method can handle the assumption violations assessed. We encourage the use of more precise scales which reduce measurement error, making the observed opinions closer to the latent continuous opinions. The alternate models we considered are of little concern except for specific cases where prediction is of primary interest. Most importantly, the inability to collect data on a full network, including all links between agents, is not a barrier to the use of this method. Instead, the inclusion of links that may or may not exist in the true network typically improves performance. While our ongoing work will continue to improve usability of this method through simulation studies to establish default hyperparameter values and investigate variability estimates, this simulation study provides researchers with the information necessary to use the method under the assumption violations expected when modeling opinion diffusion on the social networks for health behavior interventions or similar applications.

## Figures and Tables

**Figure 1 ijerph-18-13394-f001:**
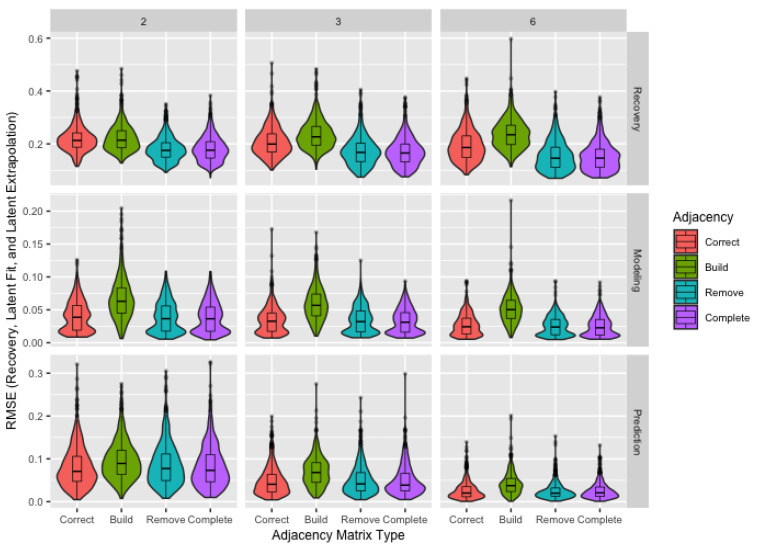
Boxplots and violin plots for root-mean-square error for recovery, modeling, and prediction by adjacency matrix type with number of time steps (horizontal) and performance metric (vertical) across facets.

**Figure 2 ijerph-18-13394-f002:**
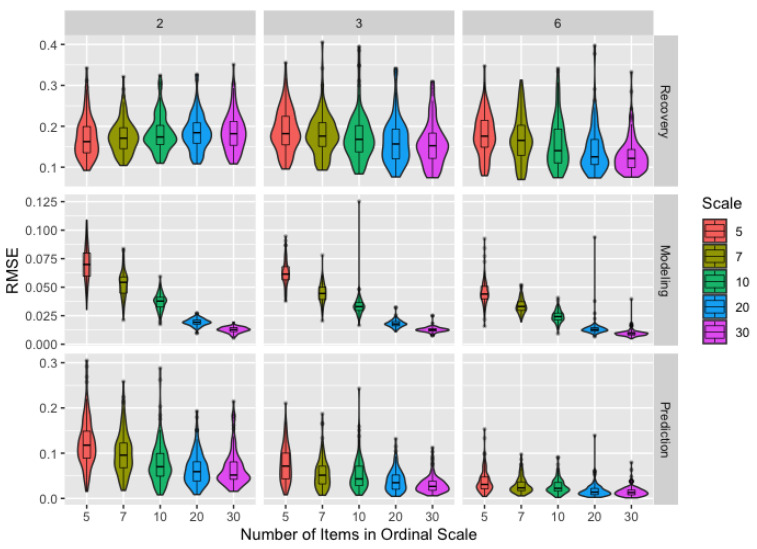
Boxplots and violin plots for root-mean-square error for recovery, modeling, and prediction by ordinal scale with number of time steps (horizontal) and performance metric (vertical) across facets for the *remove* matrix.

**Figure 3 ijerph-18-13394-f003:**
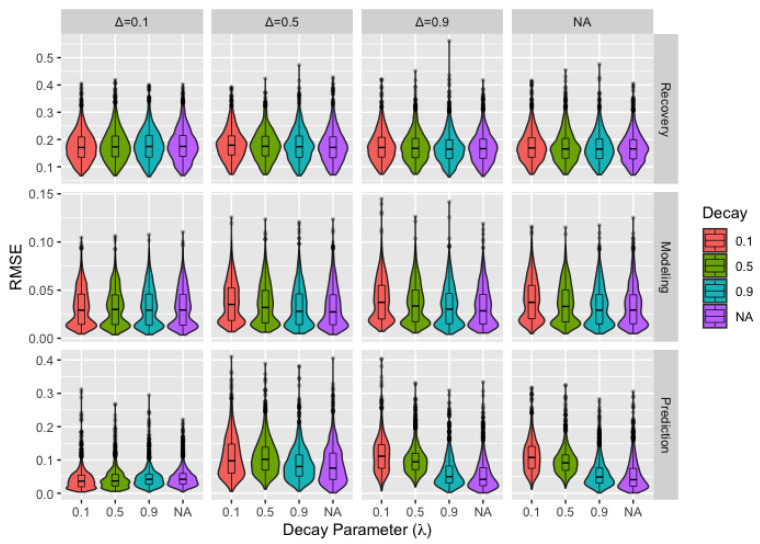
Boxplots and violin plots for root-mean-square error for recovery, modeling, and prediction by decay parameter with bounded confidence parameter (horizontal) and performance metric (vertical) across facets for the *remove* matrix.

**Figure 4 ijerph-18-13394-f004:**
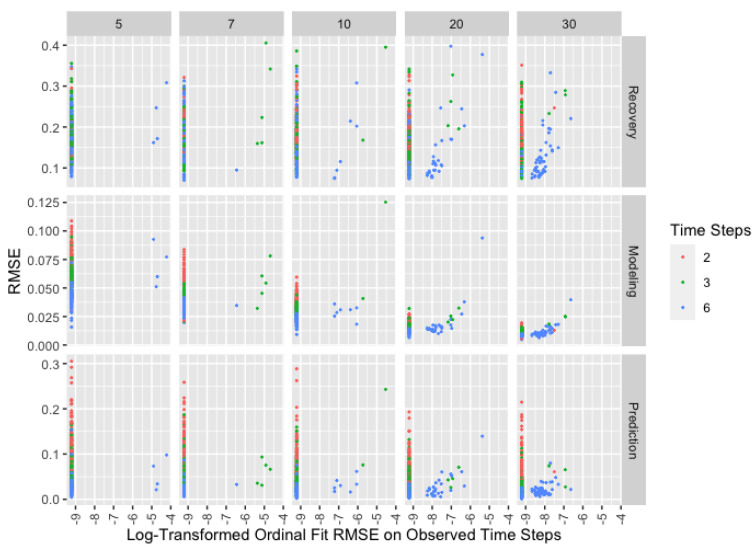
Root-mean-square error for recovery, modeling, and prediction by log-transformed RMSE for ordinal fit with shift of 0.0001 and number of time steps with number of items in ordinal scale (horizontal) and performance metric (vertical) across facets for the *remove* matrix.

**Figure 5 ijerph-18-13394-f005:**
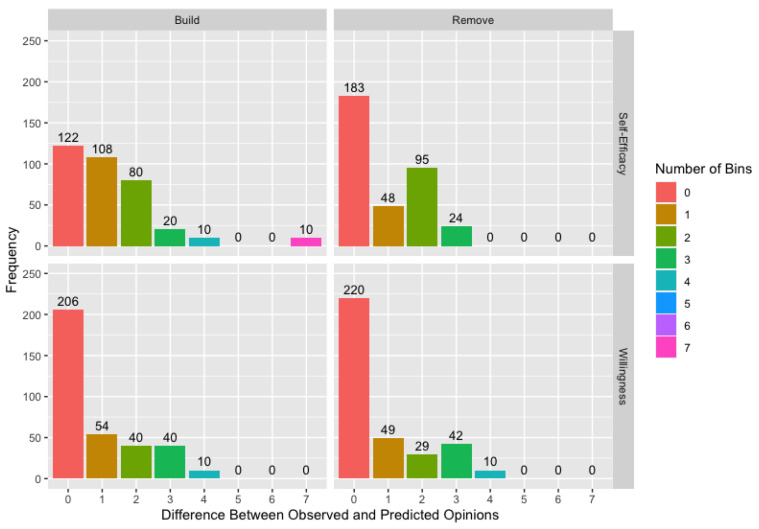
Difference between observed and modeled opinions measured in number of bins by adjacency matrix variety and measure across all networks and runs of the algorithm, originally published in Johnson et al. [2].

**Figure 6 ijerph-18-13394-f006:**
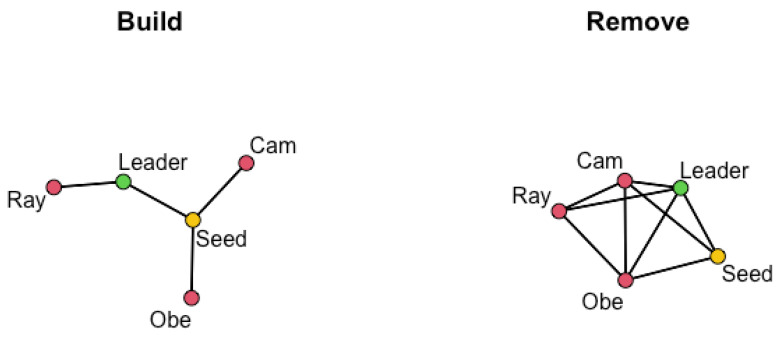
Representations of network 4 using build and remove adjacency matrices with the seed identified in yellow and the agent who attended leadership training in green.

**Table 1 ijerph-18-13394-t001:** Inputs used in the performance simulation study.

Input	Values	Notes
Network Size	N=10,20,50	Reachability enforced
Degree	d=5,9	Minimum degree d=1 for all nodes
Self-Weight	$w_{ii}=0.5$	Beta Distribution with κ=α+β=4
Time Steps	T=2,3,6	Performance assessed on t=1,...,20
Scale	n=5,7,10,20,30	
Recruitment Probability	$p_{r}=0.5,1$	
Adjacency Matrix	correct, build, remove, complete	
Bounded Confidence	Δ=0.1,0.5,0.9,1	Δ=1 equivalent to DeGroot model
Decay	λ=0.1,0.5,0.9,1	λ=1 equivalent to DeGroot model

**Table 2 ijerph-18-13394-t002:** Mean estimated weights for willingness across 10 runs for network 5 using build and remove matrices.

	Willingness							
	**Build**				**Remove**			
Eli	0.50	0.50	0.00	0.00	0.48	0.52	0.00	0.00
Jay	0.49	0.51	0	0	0.53	0.47	**0.00**	**0.00**
Uba	0.04	0	0.96	0	0.00	**0.00**	0.68	**0.32**
Max	0.21	0	0	0.79	0.12	**0.09**	**0.00**	0.79
	Eli	Jay	Uba	Max	Eli	Jay	Uba	Max

**Table 3 ijerph-18-13394-t003:** Mean estimated weights for self-efficacy across 10 runs for network 5 using build and remove matrices.

	Self-Efficacy							
	**Build**				**Remove**			
Eli	0.71	0.00	0.00	0.29	0.71	0.00	0.00	0.29
Jay	0.12	0.88	0	0	0.10	0.88	**0.00**	**0.02**
Uba	0.00	0	1.00	0	0.00	**0.00**	1.00	**0.00**
Max	0.00	0	0	1.00	0.00	**0.00**	**0.00**	1.00
	Eli	Jay	Uba	Max	Eli	Jay	Uba	Max

**Table 4 ijerph-18-13394-t004:** Mean weight placed on leaders and non-leaders and difference (leader−non-leader) by network and measure, excluding self-weight and agents without leader connections.

Network	Willingness			Self-Efficacy		
	Leader	Non-Leader	Difference	Leader	Non-Leader	Difference
1	**0.11**	0.05	0.06	0.04	**0.05**	−0.01
2	**0.09**	0.08	0.01	0.04	**0.09**	−0.05
3	**0.25**	0.11	0.14	0.02	**0.07**	−0.05
4	0.02	**0.13**	−0.11	**0.30**	0.05	0.25
5	**0.21**	0.05	0.16	0.02	**0.05**	−0.03

**Table 5 ijerph-18-13394-t005:** Mean estimated weights for willingness and self-efficacy across 10 runs for network 4 using the remove matrix.

	Willingness					Self-Efficacy				
Seed	0.40	0.01	0.57	0.02	0	0.72	0.08	0.15	0.04	0
Cam	0.18	0.70	**0.06**	**0.01**	**0.04**	0.00	0.70	**0.00**	**0.25**	**0.05**
Obe	0.00	**0.00**	0.99	**0.00**	**0.00**	0.01	**0.00**	0.99	**0.00**	**0.00**
Leader	0.00	**0.32**	**0.00**	0.68	0.00	0.00	**0.00**	**0.00**	1.00	0.00
Ray	0	**0.01**	**0.61**	0.03	0.35	0	**0.00**	**0.07**	0.91	0.02
	Seed	Cam	Obe	Leader	Ray	Seed	Cam	Obe	Leader	Ray

## Data Availability

The data generated and analysed during the simulation study are available in the file “useability.csv” in the corresponding author’s GitHub repository: https://github.com/karajohnson4/DeGrootGeneticAlgorithm (accessed on 11 November 2021). The genetic algorithm code is also available in the corresponding author’s GitHub repository under the name Algorithm-Code. The IJERPH-Archive branch will serve as an archived version. The code is written in Julia, is platform independent, requires Julia 1.5 or higher, and uses the GNU GENERAL PUBLIC LICENSE [8]. The data from the pilot study are available from cairdirector@mcw.eduJeffrey A. Kelly, PhD but restrictions apply to the availability of these data, which were used under license for the current study, and so are not publicly available. Data are, however, available from the authors upon reasonable request and with permission of cairdirector@mcw.eduJeffrey A. Kelly.

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
