# Peer review of "Performance of a Genetic Algorithm for Estimating DeGroot Opinion Diffusion Model Parameters for Health Behavior Interventions"

_ijerph, 2021, doi:10.3390/ijerph182413394_

Round 1

Reviewer 1 Report

Congratulations for your research! I think it is a very interesting topic and even though the analysis presents many limitations, it also means a very promising area of research that can bring opportunities as it presents clear practical implications that, by the way, must be reinforced in text.

Here, I offer to you some recommendations to improve your article:

In the introduction I miss, a paragraph at the end anticipating in brief what the main practical implications of the results of this research will be.

And to end the introduction, after this paragraph, I miss another one describing in brief what the main structure of the article is…Something that starts by:

"Following, after this introduction in section 2 …, section 3 presents….Finally, last section…"

In discussion part, authors should include more differences and similarities of the results found with previous research. More discussion based on similar analysis is required. More attention should be paid to what is key in results of this analysis to keep with this line of research, apart of improving the sample itself.

I miss that you describe theoretical and practical implications of this research. Authors recognize limitations, but they should also indicate future areas of research.

Author Response

In the introduction I miss, a paragraph at the end anticipating in brief what the main practical implications of the results of this research will be.

  • We included information at the end of the second and last paragraphs of the introduction to clarify the practical implications of the simulation study.

And to end the introduction, after this paragraph, I miss another one describing in brief what the main structure of the article is…Something that starts by:

"Following, after this introduction in section 2 …, section 3 presents….Finally, last section…"

  • We reworked the final paragraph of the introduction to include this.

In discussion part, authors should include more differences and similarities of the results found with previous research. More discussion based on similar analysis is required. More attention should be paid to what is key in results of this analysis to keep with this line of research, apart of improving the sample itself.

  • We added clarification in the first paragraph of section 4 that the discussion of selected PrEP results is not intended as a comprehensive analysis but as an example. We are unable to compare our method to a similar analysis since the methodological development was motivated by a lack of appropriate alternatives.

I miss that you describe theoretical and practical implications of this research. Authors recognize limitations, but they should also indicate future areas of research.

  • We added information highlighting the practical implications to the first and last paragraphs of section 5 and future work to the last paragraph.

Reviewer 2 Report

A work of quality presented with clarity and explanation.  For the non-statistics expert, the authors are invited to provide a laymen explanation to highlight the significance of the researcn methodlogy, and the applicability of the statiscal model adopted. 

Author Response

A work of quality presented with clarity and explanation.  For the non-statistics expert, the authors are invited to provide a laymen explanation to highlight the significance of the researcn methodlogy, and the applicability of the statiscal model adopted. 

  • We added information to both the second paragraph of section 1 and to section 2.1.1. to highlight the benefits of being able to fit these models, including explanations for readers with a variety of statistical backgrounds. 

Reviewer 3 Report

The authors developed a genetic algorithm to fit the DeGroot opinion diffusion model in settings with small social networks and limited follow-up of opinion change. In this article, the authors assess the algorithm performance under the less-than-ideal conditions that are likely to arise in practical applications. For this, a simulation study to assess the performance of the algorithm in the presence of ordinal opinion measurements, network sampling, and model misspecification was performed. The authors claim that the method handles alternate models well, performance depends on the precision of the ordinal scale, and sampling the full network is not necessary to use this method. On the other hand, insights were obtained from the simulation study to investigate notable features of opinion diffusion models for a social network intervention to increase the uptake of pre-exposure prophylaxis (PrEP) among Black men who have sex with men (BMSM).

The article is well written, well structured, and has relevant contributions to the field. Therefore, I have a few suggestions:

1 - What is the need to carry out this study?

2 - Why was the study conducted in 2016-2017 and not in another period? Why was the study conducted in Milwaukee and not elsewhere?

3 - Justify the sampling selection.

4 - Highlight the research's theoretical, practical, and political implications.

5 - All equations must be cited in the text.

6 - Insert suggestions for future research in the conclusions.

Author Response

1 - What is the need to carry out this study?

  • The main purpose of this paper is to present the results of the simulation study to provide users of the method information about its performance under the conditions expected in public health research. Per the recommendation of other reviewers, we clarified this point in sections 1, 4, and 5. As it serves only as an example, the specific details of the motivation, implementation, and implications of the PrEP study would not be appropriate to include here. Multiple citations are included for previous work with the PrEP data which address these concerns, including a specific reference to further information on procedures in Kelly et al. at the end of section 2.2.3.

2 - Why was the study conducted in 2016-2017 and not in another period? Why was the study conducted in Milwaukee and not elsewhere?

  • (see 1)

3 - Justify the sampling selection.

  • (see 1)

4 - Highlight the research's theoretical, practical, and political implications.

  • (see 1)

5 - All equations must be cited in the text.

  • We included references to each equation when it first appears in the text.

6 - Insert suggestions for future research in the conclusions.

  • We incorporated future work to the final sentence of section 5.